# Validity and Reliability of IPAQ-SF and GPAQ for Assessing Sedentary Behaviour in Adults in the European Union: A Systematic Review and Meta-Analysis

**DOI:** 10.3390/ijerph18094602

**Published:** 2021-04-26

**Authors:** Kaja Meh, Gregor Jurak, Maroje Sorić, Paulo Rocha, Vedrana Sember

**Affiliations:** 1Faculty of Sport, University of Ljubljana, 1000 Ljubljana, Slovenia; gregor.jurak@fsp.uni-lj.si (G.J.); maroje.soric@kif.unizg.hr (M.S.); vedrana.sember@fsp.uni-lj.si (V.S.); 2Faculty of Kinesiology, University of Zagreb, 10000 Zagreb, Croatia; 3Portuguese Institute of Sport and Youth, 1990-100 Lisbon, Portugal; paulo.rocha@ipdj.pt

**Keywords:** human activities, exercise, lifestyle, physical fitness, surveys and questionnaires, self-report, public health surveillance

## Abstract

Current lifestyles are marked by sedentary behaviour; thus, it is of great importance for policymaking to have valid and reliable tools to measure sedentary behaviour in order to combat it. Therefore, the aim of this review and meta-analysis is to critically review, assess, and compile the reliability, criterion validity, and construct validity of the single-item sedentary behaviour questions within national language versions of most commonly used international physical activity questionnaires for adults in the European Union: The International Physical Activity Questionnaire-Short Form and the Global Physical Activity Questionnaire. A total of 1749 records were screened, 287 full-text papers were read, and 14 studies were included in the meta-analysis. The results and quality of studies were evaluated by the Quality Assessment of Physical Activity Questionnaires checklist. Meta-analysis indicated moderate to high reliability (r_w_ = 0.59) and concurrent validity (r_w_ = 0.55) of national language versions of single-item sedentary behaviour questions. Criterion validity was rather low (r_w_ = 0.23) but in concordance with previous studies. The risk of bias analysis highlighted the poor reporting of methods and results, with a total bias score of 0.42. Thus, we recommend using multi-item SB questionnaires and smart trackers for providing information on SB rather than single-item sedentary behaviour questions in physical activity questionnaires.

## 1. Introduction

Sedentariness has taken over our lives in recent decades. Knowledge-based work usually requires sitting during working hours [1] and transport, leisure activities and socialisation are associated with an increase in sedentary behaviour (SB) [2]. For the purpose of this study, SB is defined as any behaviour in the waking state with an energy expenditure of ≤1.5 metabolic equivalents (METs) during a sitting or lying position [3].

Although SB and physical inactivity appear to be synonyms, there are major differences between the two terms. People who do not meet the WHO physical activity (PA) guidelines [4] are classified as physically inactive, while people who spend most of their waking hours sitting are classified as sedentary [3]. People meeting PA guidelines may be sedentary, and physically inactive people are not necessarily sedentary [2,5]. Both SB and physical inactivity have been associated with increased health risks [6,7,8,9], and light PA may be more beneficial than prolonged sitting [2]. One of the positive findings of recent studies is that high PA mitigates the negative effects of prolonged sitting, such as an increased risk of mortality [10].

SB can be measured by various methods, which are divided into subjective and objective methods similar to PA. Subjective methods rely on participants’ recollection or recording of activities [11,12], most commonly used are self-reported measures, such as questionnaires [13,14]. Objective methods measure SB (or PA) directly with wearable monitors [11] and through device-based measures, such as accelerometry [14]. In large surveys, questionnaires are typically used [13,15] because they are inexpensive, can be used with a wide range of participants [16], and include information about the setting, context, and type of SB [13]. Questionnaires can collect information exclusively on SB, or they can be a part of a PA questionnaire [5]. In addition, SB can be measured with a single item or by a composition of several questions [13]. Measuring SB with the PA questionnaires makes them easy to use in large-scale population surveys and makes it easier to gather a lot of data with one self-report instrument. On the other hand, these instruments are not designed to measure SB, which can affect the results, and the reliability and validity of SB questions in PA questionnaires are rarely tested. Device-based measures are generally considered more valid and reliable [14,17] but are based on the recording of body movements at precise times [18] or time spent in specific postures [5]. As a result, the devices lack contextual information [5] that can provide crucial information about the typical SB of each participant or group of participants (e.g., sitting at work, screen time).

Most PA questionnaires include single-item SB questions [13] and many national and international PA surveys are based on such PA questionnaires [5]. Therefore, the reliability and validity of these specific single-item SB questions are essential for an accurate and precise assessment of sedentariness in society. In a recent meta-analysis based on 96 studies, the validity of SB questions was found to be moderate to low [19], which is consistent with previous findings [5,14]. Studies focusing on instruments with single items measuring SB also reported low validity (Spearman ρ ranging from 0.012 to 0.46), with insufficient and non-detailed information on comparability with sitting time measured with accelerometers [20,21,22]. The reliability of instruments assessing SB tends to be higher than validity, moderate to high [5,19,20,21], and regularly performed behaviours (sitting at work, watching TV) tend to have higher reliability coefficients [14]. There are statistically significant differences in criterion validity of single-item SB questions compared to multi-item questions and activity logs, and a recent meta-analysis reported between-group differences with higher validity for multi-item instruments [5]. Conversely, another meta-analysis found no differences in correlation coefficients between single-item and multi-item SB questions [19], highlighting the need for further validity testing of single-item SB questionnaires. As none of the previous meta-analyses focused exclusively on single-item SB questions, that are part of an international PA questionnaire, there is a need for further research. In the context of the EU, only single-item SB questions are used to measure SB, therefore the reliability and validity of these questions should be tested and analysed.

The EUPASMOS project aims to create a unified framework for measuring PA, SB, and sport participation in the European Union (EU) member states. In this meta-analysis, we included studies on the measurement properties of the three most common international PA questionnaires used for PA surveillance conducted in the EU. Only studies that used official national language versions of the selected questionnaires were included. Studies met the inclusion criteria if they were based on one of the three most frequently used PA questionnaires with a single-item SB question in the EU described elsewhere [23]. All selected PA questionnaires: International Physical Activity Questionnaire-Short Form (IPAQ-SF) [24], Global Physical Activity Questionnaire (GPAQ) [25,26], and European Health Interview Survey-Physical Activity Questionnaire (EHIS-PAQ) [27,28], contain a single-item SB question. Although the questionnaires are similar, and none of them measures dimensions or type of SB, the main difference is the recall period. IPAQ-SF contains a question about sitting time on a workday during the last seven days (participants indicate hours and minutes) and is the first internationally recognised PA surveillance instrument [29,30], and GPAQ contains a question about usual sitting time on a workday (participants indicate hours and minutes) and is the most widely used PA questionnaire in the world, with the use in 120 countries [29,30]. Lastly, EHIS-PAQ contains a question for the usual week of sitting and lying down during the day (multiple choice question) [28]. The reliability and validity of both IPAQ-SF and GPAQ SB questions differ significantly. The main differences for all three constructs may be seen for the IPAQ-SF (e.g., concurrent validity: Spearman ρ = 0.23–0.97 [20,31,32,33]). For the GPAQ, smaller differences are found for all three constructs, but results differ between studies (e.g., concurrent validity: Spearman ρ ranging from 0.56 to 0.89) [25,34]. Several systematic reviews and meta-analyses of the reliability and validity of SB measurement have been published in recent years [5,13,19], but there is no research focusing on single-item SB questions, although most surveillance systems for measuring and reporting SB in society are based on single-item questions. Moreover, there is a lack of evidence regarding measurement properties of single-item SB questions in the EU, while the same questionnaires (and hence the same SB questions) are used for national and European policymaking to combat SB, as well as for cross-national comparisons (e.g., Eurobarometer). It is therefore important to examine the validity and reliability of the data gathered using single-item SB questions.

Thus, this systematic review and meta-analysis aims to critically review, compile, and assess the reliability, criterion validity, and construct validity of the single-item SB questions within national language versions of the most commonly used international PA questionnaires in the EU.

## 2. Materials and Methods

The present systematic review and meta-analysis were performed according to a protocol designed a priori following recommendations set by the Preferred Reporting Items for Systematic Reviews and Meta-Analyses (PRISMA) guidelines [35,36]. The present work has been registered at the International Prospective Register for Systematic Reviews, identification code CRD42020138845.

### 2.1. Search Strategy

We first searched PubMed, SportDiscus, and Scopus databases for studies reporting measurement characteristics of IPAQ-SF, GPAQ, and EHIS-PAQ in April and May 2018. Subsequently, the Dart, ResearchGate, and Google Scholar databases were also searched. An identical search string was repeated in May 2020 to include articles published between May 2018 and May 2020. We used the following search string: “the name of the questionnaire (e.g., IPAQ) AND (valid * OR reliab * OR repeat * OR reproducib * OR assess * OR measure *).” Additional studies were identified by searching the reference lists of the full papers that met the eligibility criteria. Grey literature was additionally screened through ResearchGate, Google Scholar, and Mendeley, using only the keyword “name of the questionnaire, e.g., GPAQ AND valid*”. Additional literature that met the eligibility criteria of the present review was also obtained through an online questionnaire published on the platform 1KA (University of Ljubljana, Faculty of Social Sciences) with the help of WHO in the framework of the EUPASMOS project activities. National focal points for health-enhancing physical activity (HEPA) were asked to report any national research, reports, and doctoral theses published in their national languages that examined the measurement properties of IPAQ-SF or GPAQ. All articles generated from the initial search were stored on the Mendeley reference management software and researcher network (Elsevier, Amsterdam, The Netherlands), which was used to remove duplicate references. Even though we performed the database search for EHIS-PAQ, we were unable to include it in the analysis as we did not identify suitable research papers.

### 2.2. Eligibility Criteria

All included studies were peer-reviewed, included healthy adults (>18 years, special populations (i.e., participants) were excluded) and were carried out in one of the European countries (28 countries were included, the United Kingdom was still part of the EU and therefore included in this review). Two independent reviewers (V.S. and K.M.) performed eligibility screening and included studies published in one of the 24 official languages of European Union. The included studies needed to report modes of administration, translation protocols, and coefficients for reliability, concurrent validity, and criterion validity.

### 2.3. Quality and Risk of Bias Assessment

Methodological quality and risk of bias assessment were performed by two independent reviewers (V.S. and K.M.). The methodological quality of the included articles included was examined using the Quality Assessment of Physical Activity Questionnaire checklist [37], which was developed specifically for assessing methodological quality in systematic reviews of PA questionnaires. The risk of bias assessment of the final sample of 14 included articles was conducted, following the criteria previously set by Sneck [38] and Sember [23,39], following Cochrane’s guidelines for Systematic Reviews and Interventions [40].

### 2.4. Data Extraction and Statistical Analysis

Systematic searches, article screening, and data extraction were performed by two independent reviewers (V.S. and K.M.) who extracted reliability and validity coefficients for single-item SB questions. In case of ambiguity, a third reviewer (G.J.) reviewed the article. In addition, senior investigators (G.J. and P.R.) checked the summary tables of the entered data and checked for any discrepancies in the data. Meta-analysis was performed following the Hunter-Schmidt approach [41], explained in more detail elsewhere [23]. Credibility intervals and I^2^ and Q statistics were calculated to measure the heterogeneity of effect size. The forest plots were generated with the help of MS Excel.

### 2.5. Grading the Level of Evidence

The levels of evidence were formulated using criteria proposed by van Poppel and colleagues [30] (Table 1). Regarding reliability levels of evidence, Pearson and Spearman correlations were considered insufficient due to known systematic errors [42]; therefore, only ICC, Kappa, or Concordance were classified as the highest level (1) of evidence. The highest level of evidence for criterion validity would be the comparison of the PA questionnaires with the gold standard: doubly labelled water (DLW) [43]. However, DLW also includes basal metabolic rate and thermal effects of food; therefore, the use of other validated instruments is more reliable for obtaining construct validity. Thus, we assessed the construct validity by analysing studies that compared one PA questionnaire with another PA questionnaire (concurrent validity), or to accelerometers (criterion validity).

## 3. Results

Figure 1 shows the flow of the study selection process. Of the 4716 studies identified through the search, 14 studies [20,21,25,31,32,33,34,44,45,46,47,48,49,50] were included in the present systematic review and meta-analysis (Figure 1). General characteristics, such as the country where the study was conducted, population, and modes and means of administration, are presented in Table 2. We included studies from 18 different EU countries, mainly from the United Kingdom (5) and Spain (3). Four studies were cross-national [20,21,46,47].

A total of 5294 people participated in the selected studies. The age range of participants included in all studies was between 18 and 75 years. In 13 of 14 studies, the participants’ gender ratio was included (93%), with the gender ratio unknown in one study [47]. In terms of sampling, nine studies used convenient (64%), three random (21%), and one multistage stratified probability sample (7.5%), while one study did not provide sample description (7.5%) [44]. Most studies (*n* = 7.5%) used a self-administered mode of administration, four studies used interviews (29%), two studies used a self-administered mode and an interview (14%), and one used a telephone interview (7%).

Table 3 and Table 4 show results on test–retest reliability, concurrent validity, and criterion validity for SB in selected studies from the EU, including study population, measured construct (sitting or SB), comparison method (type of accelerometer or PA questionnaires), results (Spearman, Pearson, ICC), and methodological quality grade (1− to 3+).

Even though only five studies assessed test–retest reliability [20,21,32,34,47], the most associations (53) were found for this measurement construct. There was a considerable variability in the reported data (Figure 2) and the weighted mean for test–retest reliability amounted to r_r_ = 0.59 (95% CI = 0.55 to 0.63). Only one study [34] was graded with the highest methodological grade, twenty-three associations were graded with grade 2, and thirty associations with the lowest methodological grade 3. Five studies assessed concurrent validity [20,25,33,34,50] and reported eighteen different associations. The range of results was considerable, similar to reliability (Figure 3), and the weighted average mean for concurrent validity was found to be r_c_ = 0.55 (95% CI = 0.42 to 0.68). Most of the associations that assessed concurrent validity were graded with grade 1, four of them were graded with grade 2, and three associations were graded with methodological grade 3. Criterion validity was observed in 10 of 13 included studies, with 24 associations for this measurement construct [20,21,31,32,34,44,45,46,48]. One result with negative criterion validity stood out (r = −0.48), otherwise criterion validity results were low but close to each other (Figure 4). The weighted average mean for criterion validity was r_cr_ = 0.23 (95% CI = 0.19 to 0.27). All associations for criterion validity were given the highest methodological grade: 3 (Table 4).

We included the qualitative assessment of the studies as a moderator in the analysis. For criterion validity, all studies were graded with grade 3; thus, no moderating effects were found. High heterogeneity of these studies due to differences in experimental design, analysis, poor study quality, etc. (Q = 10, *p* = 0.33), could also be the reason for the low results and grading (Table 5). For reliability and concurrent validity, we detected the moderating effects of the quality assessment. Each of the three grades were moderators of reliability and concurrent validity with statistically significant results (*p* < 0.00 for both constructs).

The Egger’s bias test [51] showed no publication bias for any of the measurement characteristics regarding SB (*p* > 0.30 for all) (Table 4). The results of the risk-of-bias assessment are presented in Table 6. The total average risk of bias for all included studies was moderate (0.42). Of the fourteen studies, only one was rated as having a low risk of bias (≥67% of total score) with a score of 0.78 of the total score, six were rated as having a moderate risk of bias (>33 and <67% of the total score) with an average of 0.48 of the total score, and seven studies were rated as having a high risk of bias (<33% of total score) with an average of 0.31 of the total score. In addition, only four studies (29%) reported power calculations to determine sufficient sample size [52].

## 4. Discussion

This systematic review and meta-analysis examined test–retest reliability, and concurrent and criterion validity of the single-item SB question in national language versions of the two most commonly used PA questionnaires in the EU: IPAQ-SF and GPAQ. Through the literature review, we identified 14 studies that adequately tested selected PA questionnaires over the last 17-year period between 2003 and 2020.

The main findings of our study are: (i) IPAQ-SF and GPAQ were criterion-validated for SB in only 10 EU countries, with low weighted average mean r_cr_ = 0.23 (95% CI 0.19 to 0.27), (ii) reporting of study methods and results was rather poor, with only one study with low risk of bias and 13 studies with moderate or high risk of bias, resulting in an overall moderate risk of bias with a total score of 0.42, and (iii) the representation of the different EU countries might be biased, as less than half of the EU countries were included in SB measurement characteristics studies.

Our research highlighted that SB is not reported in all studies using PA questionnaires, although all of these studies collected data on SB. Thirteen EU countries were represented in 14 studies (6 from the UK, 3 from Spain and the Netherlands, 2 from Belgium, Finland, France, Germany, Portugal, and Sweden, and one from Austria, Italy, Lithuania, and Slovenia). Furthermore, four of the studies were conducted internationally [20,21,46,47], two of them with small sample sizes [21,46], and one study did not report the results separately for each participating country [46]. Therefore, the representation of the different EU countries may be biased. We can assume that PA is still considered as a more relevant factor or contributor to a healthy lifestyle, even though studies in recent years have highlighted the effect physical inactivity and SB have on the health [2,53], and the 24 h movement behaviour approach advocates for the integration of all movement behaviours across the 24 h daily span in the movement guidelines [54].

Furthermore, single-item SB questions in IPAQ-SF and GPAQ were criterion-validated in only 10 EU countries (Belgium, Finland, France, Germany, Lithuania, the Netherlands, Slovenia, Spain, Sweden, and the United Kingdom). As expected, the results for criterion validity are low but still perplexing, with coefficients r_w_ = 0.19 to 0.27, similar to the low correlation coefficients for MVPA presented in the previous meta-analysis on the same PA questionnaires [23]. Similarly, startling results were shown in the recent meta-analysis, where both IPAQ-SF and GPAQ results showed underreporting of SB compared to accelerometers (mean difference for IPAQ-SF = −161.67 and for GPAQ = −219.85), whereas multi-item SB questionnaires performed much better (e.g., mean difference for SBQ = −5.8 and Marshall sitting questionnaire = 83.85) [5]. Poor validity results of MVPA and SB are even more concerning when recommendations and PA-enhancing measures are based on them. Some previous validation studies using the same PA questionnaires on non-European populations came to similar conclusions [5,14], while studies including multiple questions on SB reported higher correlations [19]. Comparison of single-item SB questions in PA questionnaires and multiple-item SB questionnaires from previous meta-analyses demonstrated advantages of the latter (better suited for the precise measurement of SB), as single-item questions significantly underreport sitting time compared to multi-item questionnaires [5,14] and are more domain-specific [19]. Multi-item SB questionnaires can help participants to remember more of their SB (e.g., transport-related SB, sitting at work) and provide more detailed information for researchers and healthcare providers [10,55].

Therefore, multi-item questionnaires can improve the criterion validity, since they are more complex [19]. Although test–retest reliability and concurrent validity are moderate to high, we aim for high criterion validity as it is the measure that compares the subjective method—the SB questionnaire—with the objective measure. Therefore, for more valid and reliable results, we propose to use questionnaires with multiple-item SB questions and to include the results in the 24 h movement behaviour guidelines. This information would provide policymakers with more valid data about the time spent sitting and the context of SB.

Results obtained using the questions of SB need to be critically evaluated and not taken for granted, as the results are likely to underestimate (or overestimate) the time spent sitting [44,45,50]. In our review, 71% of the included studies in 10 countries (Belgium, Germany, Finland, France, Lithuania, the Netherlands, Slovenia, Spain, Sweden, UK) also analysed the data for criterion validity. Although the percentage of studies is high, only one third of the EU countries conducted a validation study using objective methods on their national version of the PA questionnaire. Without an acceptably validated national language version of the most commonly used PA questionnaires and corresponding SB question, it is impossible to compare data between countries in the EU or to base health programs even on the results of inadequate questionnaires [56,57,58,59].

As expected, the test–retest reliability of single-item SB questions was moderate to high (r_w_ = 0.552 to 0.63), which is similar to the reliability of PA constructs in the same PA questionnaires (e.g., moderate PA r_w_ = 0.37 to 0.43 and vigorous PA r_w_ = 0.49 to 0.58) [23]. The test–retest interval should be between three and eight days [52], and the recall periods of all included studies were within the desired range and consistent with previous studies [5,14,19]. Moreover, the concurrent validity of single-item SB questions was found to be moderate to high, with a weighted mean ranging from 0.42 to 0.68. The weighted mean of moderate to vigorous PA from a recent meta-analysis of the same PA questionnaires was lower, compared to SB (r_w_ MVPA = 0.36 to 0.46) [23]. As highlighted in a meta-analysis by Bakker and colleagues [19], measuring subjective perceptions of physical (in)activity behaviours, from sedentariness to vigorous PA, is complicated; therefore, validity results are moderate.

The analysis of risk of bias highlighted poor reporting of methods and results, with a total bias score of 0.42. Only 14 of 110 SB measurement characteristics received grade 1, 28 were rated with grade 2, and 68 with grade 3. In particular, criterion validity received low grades due to low correlation coefficients between objective and subjective methods, and all 24 criterion validity constructs were given the lowest grade 3. Low qualitative ratings were usually awarded because the studies did not use the interclass correlation (ICC), Kappa, or Concordance reliability score, but instead mostly used Spearman coefficient of association.

We conclude that EU studies validating SB questions in IPAQ-SF or GPAQ are methodologically inadequate, which is in line with the meta-analysis conducted on PA and on the same PA questionnaires in the EU [23]. This is a cause for concern, as more than half of the constructs did not follow the preferred recommendations for the assessment or the reliability and validity of PA questionnaires, and this requires a more rigorous study design in future reliability and validity studies. We recommend that researchers use Kappa or ICC, as these also take the rater bias into account [60].

### Strengths and Limitations

Eurobarometer, as a cross-national comparison in the EU, and most of the national surveillance systems for measuring and reporting SB, are based on single-item SB questions. Our research examined several available sources to find validation studies on the three most common PA questionnaires in EU national languages. However, the present systematic review and meta-analysis has several limitations that should be considered when extrapolating the results: (i) first, the review was limited to only two of the most commonly used PA questionnaires in the EU, which include only single-item SB questions, (ii) although we searched five databases, it is possible that not all relevant studies are included in the present review, (iii) not all questionnaires and studies were conducted in the same week or period of the year, leading to potentially different results due to different weather conditions in the EU and yielding different results of the reported SB, and (iv) one of the studies reported SB for weekdays and weekends, which is not consistent with the IPAQ-SF protocol [50]. (v) Although the quality of each study was assessed, the results for criterion validity did not differ, as all studies were graded with the lowest grade, which also influenced the high heterogeneity of said construct. (vi) Coefficients of association were reported regardless of whether they were significant or insignificant in the initial studies, which could lead to different results if only significant results were used. (vii) This review includes studies from the UK, although, the UK is no longer a part of the EU at the time of publication, and (viii) the results of the present meta-analysis refer only to the adult population and are not necessarily valid for other populations, such as the elderly, children, and patients.

## 5. Conclusions

The single-item SB questions in the EU national versions of GPAQ and IPAQ-SF have low criterion validity; therefore, the assessment of SB in EU populations based on these PA questionnaires is critical. Policymakers should be aware of the limitations of information on SB collected with the aforementioned PA questionnaires when evaluating policies. We recommend the use of commercial fitness trackers to monitor sedentary behaviour within the 24 h movement behaviour paradigm in combination with multi-item SB questionnaires.

Such a combination could provide more valid and non-expensive information on the time and context of SB necessary to determine behavioural interventions. Additionally, the surveillance of physical fitness as a measure of health (PA, physical inactivity, and SB influence physical fitness) should be considered as a much better alternative to PA and SB questionnaires. Finally, researchers should report the results of SB questions in a standardised manner to improve the quality of the assessment and reduce the risk of bias.

## Figures and Tables

**Figure 1 ijerph-18-04602-f001:**
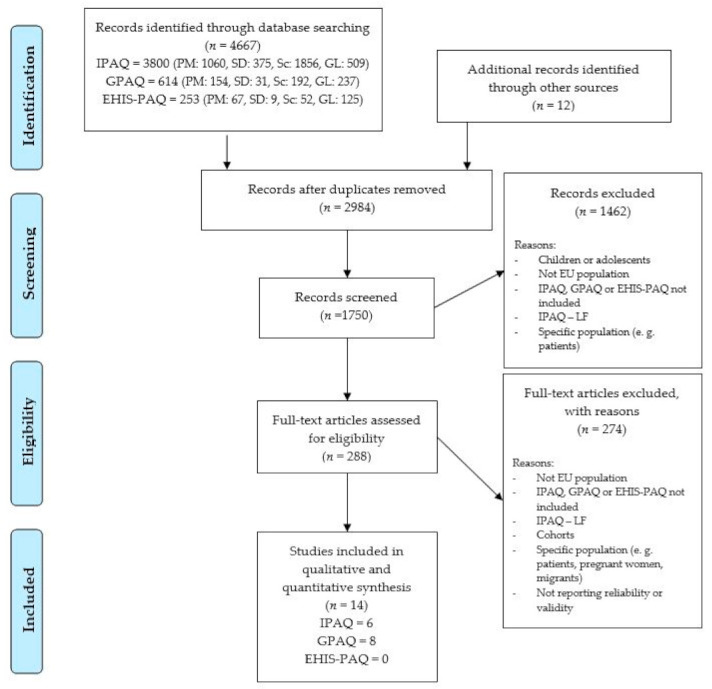
Flowchart showing study identification process. PM—PubMed; SD—SportDiscus; Sc—Scopus; GL—grey literature.

**Figure 2 ijerph-18-04602-f002:**
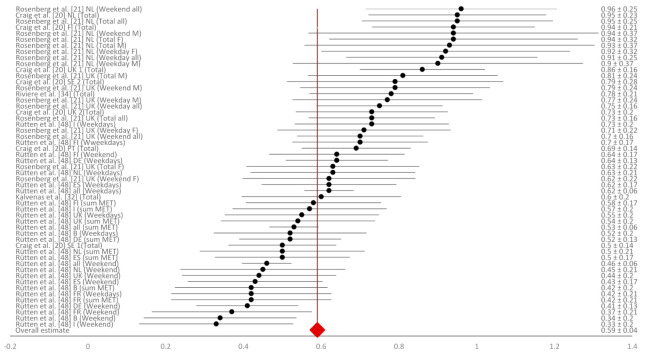
Forest plot of reliability coefficients with 95% confidence intervals. Notes: B—Belgium; DE—Germany; ES—Spain; FI—Finland; FR—France; I—Italy; LT—Lithuania; NL—The Netherlands; PT—Portugal; SE—Sweden; UK—United Kingdom; M—male; F—female.

**Figure 3 ijerph-18-04602-f003:**
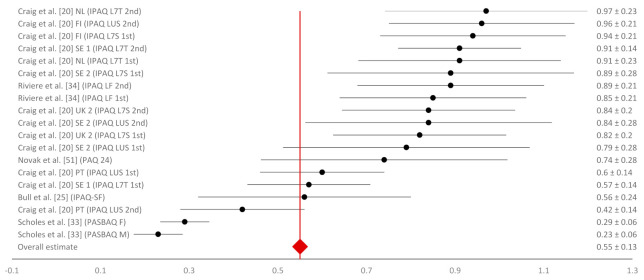
Forest plot of concurrent validity coefficients with 95% confidence intervals. Notes: B—Belgium; FI—Finland; FR—France; NL—The Netherlands; PT—Portugal; SE—Sweden; UK—United Kingdom; PAQ 24—Physical Activity Questionnaire for 24 h; PASBAQ—Physical Activity and Sedentary Behaviour Assessment Questionnaire; M—male; F—female.

**Figure 4 ijerph-18-04602-f004:**
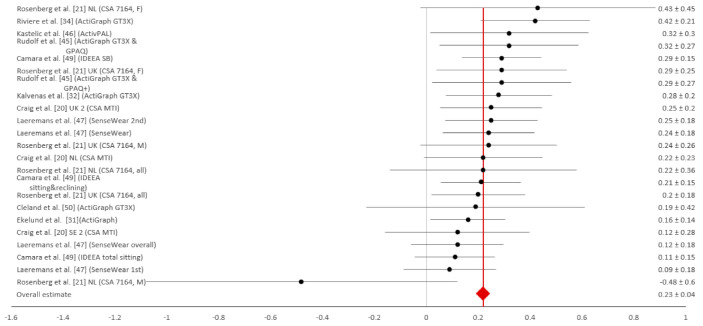
Forest plot of criterion validity coefficients with 95% confidence intervals. Notes: B—Belgium; FI—Finland; NL—The Netherlands; PT—Portugal; SE—Sweden; UK—United Kingdom; SB—sedentary behaviour; IDEAA— Intelligent Device for Energy Expenditure and Activity; M—male; F—female.

**Table 1 ijerph-18-04602-t001:** Levels of evidence.

Level of Evidence	Reliability	Concurrent Validity	Criterion Validity
1	Adequate time between test–retest and use of ICC, Kappa or Concordance reliability score > 0.70.	Comparison method to other PA questionnaires and concurrent validity score > 0.8.	Comparison method to accelerometers or doubly, calorimetry or doubly labelled water, criterion validity score > 0.8.
2	Inadequate time between test–retest and use of ICC, Kappa or Concordance reliability score < 0.70; adequate time between test-retest and use of Pearson or Spearman correlation < 0.7.	Comparison method to other PA questionnaires; 0.8 > concurrent validity score > 0.5.	Comparison method to accelerometers or doubly, calorimetry or doubly labelled water; 0.8 > validity score > 0.5.
3	Inadequate time between test–retest and use of Pearson or Spearman correlation < 0.70.	Comparison method to other PA questionnaires; concurrent validity score < 0.50.	Comparison method to accelerometers or doubly, calorimetry or doubly labelled water; criterion validity score < 0.50.
Positive (+) score	Studies with more than 50 participants and reliability of coefficients > 0.70.	Studies with more than 50 participants.	Studies with more than 50 participants.
Negative (−) score	Studies with less than 50 participants and reliability coefficients < 0.70.	Studies with less than 50 participants.	Studies with less than 50 participants.

**Table 2 ijerph-18-04602-t002:** General characteristics of selected studies assessing SB.

Author (PA Questionnaire) Language Version	Country	Population **				
		Size	Age; (Range)	Gender (Male, Female)	Sample Description	Mode and Means of Administration
Bull et al. [25] (GPAQ), Portuguese	PT	67	18–75	17 + 50	Prevalence of young participants (18–44, *n* = 56). Convenient regional sample.	Interview. Unknown mode.
Cámara et al. [48] (GPAQ), Spanish	ES	163	70 (67–75)	67 + 96	Older adults from IMPACT65+ study.	Interview. Face to face.
Cleland et al. [49] (GPAQ), English	UK	22	46	8 + 14	Random national sample.	Self-administered. Unknown mode.
Craig et al. [20] (IPAQ-SF), German, English, Finnish, Dutch, Portuguese, Swedish	Cross-national: AT, UK, FI, NL, PT, SE	2115:	47	77 + 123	Specific populations. Convenient samples, but collectively, the participants represented a wide range of age, education, income, and activity levels.	Self-administered. Unknown modes.
		200 SE1; 50 SE2;	41	22 + 28		
		149 UK1	35	68 + 81		
		101 UK2	41	38 + 63		
		88 FI	56	43 + 45		
		196 PT	35	96 + 100		
		74 NL	33	34 + 40		
Ekelund et al. [31] (IPAQ-SF), Swedish	SE	185	42 (20–69)	93 + 92	Workers and students. Convenient regional sample.	Telephone interview.
Kalvenas et al. [32] (IPAQ-SF), Lithuanian	LT	92 ^#^	18–69	reliability 29 + 63 validity 23 + 58	Employees of university and private company. Convenient sample from urban area.	Self-administered. Unknown mode.
Kastelic et al. [5] (GPAQ), Slovenian	SI	42	M 39 F 50	37 + 5	Crane operators and office workers. Convenient sample.	Interview. Unknown mode.
Laeremans et al. [46] (GPAQ), German, Spanish, English	Cross-national: B. ES, UK	122: 41 B; 41 ES; 40 UK	35	55 + 67	Random regional sample.	Self-administered. Online.
Novak et al. [50] (GPAQ), German	AT	50	25	39 + 11	Students. Convenient sample.	Self-administered. Unknown mode.
Rivière et al. [34] (GPAQ), French	FR	87 ^###^	30	25 + 67	Medical personnel and students. Convenient sample.	Interview and self-administered. Unknown mode.
Rosenberg et al. [21] (IPAQ-SF), English, Dutch	UK, NL	UK 146 (118) * NL 64 (30) *	UK 35.3 NL 32.7	UK 65 + 78 NL 28 + 38 (UK 56 + 61 NL 11 + 19) *	Convenient sample. University staff and students.	Self-administered. Unknown mode.
Rudolf et al. [44] (GPAQ), German	DE	54	28	23 + 31	University students. Convenient sample.	Self- administered. Online.
Rütten et al. [47] (IPAQ-SF), German, Finnish, French, Italian, Dutch, Spanish, English	Cross-national: B, FI, FR, DE, I, NL, ES, UK	951: 100 B; 127 FI; 91 FR; 223 DE; 98 I; 86 NL; 128 ES; 98 UK	>18	Unknown	Random sample.	Interview. Face to face.
Scholes et al. [33] (IPAQ-SF), English	UK	1252	>16	548 + 704	Multistage stratified probability sampling.	Self-administered. Pen and paper.

Notes: All included articles are presenting frequency and duration as sitting parameters; AT—Austria; B—Belgium; DE—Germany; ES—Spain; FI—Finland; FR—France; I—Italy; LT—Lithuania; NL—The Netherlands; PT—Portugal; SI—Slovenia; UK—United Kingdom; VPA—vigorous PA; MPA—moderate-to-vigorous PA; LPA—light PA; ^#^ 92 reliability and 81 validity; ^###^ 68 reliability and 87 criterion validity; *—sample size for validity study; ** population (size, age, gender) was presented only for European country, nevertheless comparisons were made cross-national.

**Table 3 ijerph-18-04602-t003:** Results for test–retest reliability for SB in selected studies.

Reference (PA Questionnaire)	Study Pop.	Construct (Comparison PA Questionnaire or Device)	Results *	Rating
Craig et al. [20] (IPAQ-SF)	SE 1	sitting	0.50	3+
	UK1	sitting	0.86	2+
	UK2	sitting	0.73	3+
	FI	sitting	0.94	2+
	PT	sitting	0.69	3+
	SE 2	sitting	0.79	2+
	NL	sitting	0.95	2+
Kalvenas et al. [32] (IPAQ-SF)	LT	sitting (min/weekday/week)	0.60	3+
Riviere et al. [34] (GPAQ)	FR	sitting	0.78 ICC = 0.80	2+ 1+
Rosenberg et al. [21] (IPAQ-SF)	UK	total sitting (male)	0.81	2+
		weekday sitting (male)	0.79	2+
		weekend sitting (male)	0.77	2+
		total sitting (female)	0.63	3+
		weekday sitting (female)	0.62	3+
		weekend sitting (female)	0.71	2+
		total sitting (all)	0.73	2+
		weekday sitting (all)	0.70	2+
		weekend sitting (all)	0.75	2+
	NL	total sitting (male)	0.93	2−
		weekday sitting (male)	0.94	2−
		weekend sitting (male)	0.90	2−
		total sitting (female)	0.94	2−
		weekday sitting (female)	0.96	2−
		weekend sitting (female)	0.92	2−
		total sitting (all)	0.95	2−
		weekday sitting (all)	0.96	2−
		weekend sitting (all)	0.91	2−
Rütten et al. [47] (IPAQ-SF)	B	sitting weekdays—total minutes	0.52	3+
		sitting weekend-total minutes	0.34	3+
		sitting sum MET	0.42	3+
	FI	sitting weekdays—total minutes	0.70	2+
		sitting weekend-total minutes	0.64	3+
		sitting sum MET	0.58	3+
	FR	sitting weekdays—total minutes	0.42	3+
		sitting weekend-total minutes	0.37	3+
		sitting sum MET	0.42	3+
	DE	sitting weekdays—total minutes	0.64	2+
		sitting weekend-total minutes	0.41	3+
		sitting sum MET	0.52	3+
	I	sitting weekdays—total minutes	0.73	2+
		sitting weekend-total minutes	0.33	3+
		sitting sum MET	0.57	3+
	NL	sitting weekdays—total minutes	0.63	3+
		sitting weekend-total minutes	0.45	3+
		sitting sum MET	0.50	3+
	ES	sitting weekdays—total minutes	0.62	3+
		sitting weekend-total minutes	0.43	3+
		sitting sum MET	0.50	3+
	UK	sitting weekdays—total minutes	0.55	3+
		sitting weekend-total minutes	0.44	3+
		sitting sum MET	0.54	3+
	All nations	sitting weekdays—total minutes	0.62	3+
		sitting weekend-total minutes	0.46	3+
		sitting sum MET	0.53	3+

Notes: * Spearman ρ, unless otherwise labelled; B—Belgium; DE—Germany; ES—Spain; FI—Finland; FR—France; I—Italy; LT—Lithuania; NL—The Netherlands; PT—Portugal; SE—Sweden; UK—United Kingdom.

**Table 4 ijerph-18-04602-t004:** Results for concurrent and criterion validity for SB in selected studies.

Reference (PA Questionnaire)	Study Pop.	Method	Construct (Comparison PA Questionnaire or Device)	Results *	Rating
Bull et al. [25] (GPAQ)	PT	CCV	sitting (IPAQ-SF)	0.56	2+
Camara et al. [48] (GPAQ)	ES	CRV	Total sitting (IDEEA)	0.11	3+
			sitting and reclining (IDEEA)	0.21	3+
			SB (IDEEA)	0.29	3+
Cleland et al. [49] (GPAQ)	UK	CRV	sitting (ActiGraph GT3X)	0.19	3−
Craig et al. [20] (IPAQ-SF)	SE1	CCV	sitting 1st session (IPAQ L7T)	0.57	2+
			sitting 2nd session (IPAQ L7T)	0.91	1+
	UK2	CRV	sitting (CSA motion detector MTI)	0.25	3+
		CCV	sitting 1st session (IPAQ L7S)	0.82	1+
			sitting 2nd session (IPAQ L7S)	0.84	1+
	FI	CRV	sitting (CSA motion detector MTI)	0.46	3+
		CCV	sitting 1st session (IPAQ LUS)	0.96	1+
			sitting 2nd session (IPAQ LUS)	0.96	1+
			sitting 1st session (IPAQ L7S)	0.94	1+
	PT	CCV	sitting 1st session (IPAQ LUS)	0.60	2+
			sitting 2nd session (IPAQ LUS)	0.42	3+
	SE 2	CRV	sitting (CSA motion detector MTI)	0.12	3+
		CCV	sitting 1st session (IPAQ LUS)	0.79	1+
			sitting 2nd session (IPAQ LUS)	0.84	1+
			sitting (IPAQ L7S)	0.89	1+
	NL	CRV	sitting (CSA motion detector MTI)	0.22	3+
		CCV	sitting 1st session (IPAQ L7T)	0.91	1+
			sitting 2nd session (IPAQ L7T)	0.97	1+
Ekelund et al. [31] (IPAQ-SF)	SE	CRV	sitting(ActiGraph)	Pearson r = 0.16	3+
Kalvenas et al. [32] (IPAQ-SF)	LT	CRV	sitting (ActiGraph GT3X)	0.28	3+
Kastelic et al. [45] (GPAQ)	SI	CRV	sitting (ActivPAL)	0.32 ICC = 0.21	3− 3−
Laeremans et al. [46] (GPAQ)	B, ES, UK	CRV	sitting 1st session (Bodymedia Fit SenseWear)	0.09	3−
			sitting 2nd session (Bodymedia Fit SenseWear)	0.25	3−
			sitting 3rd session (Bodymedia Fit SenseWear)	0.24	3−
			Overall sitting (Bodymedia Fit SenseWear)	0.12	3−
Novak et al. [50] (GPAQ)	AT	CCV	sitting (PAQ 24)	0.74	2+
Riviere et al. [34] (GPAQ)	FR	CRV	sitting (ActiGraph GT3X)	0.42	3+
		CCV	sitting 1st session (IPAQ-LF)	0.85	1+
			sitting 2nd session (IPAQ-LF)	0.89	1+
Rosenberg et al. [21] (IPAQ-SF)	UK	CRV	sitting (male, CSA 7164)	0.24	3+
			sitting (female, CSA 7164)	0.29	3+
			sitting (all, CSA 7164)	0.20	3+
	NL	CRV	sitting (male, CSA 7164)	−0.48	3−
			sitting (female, CSA 7164)	0.43	3−
			sitting (all, CSA 7164)	0.22	3−
Rudolf et al. [44] (GPAQ)	DE	CRV	sitting (ActiGraph GT3X and GPAQ +)	0.32	3+
			sitting (ActiGraph GT3X and GPAQ)	0.29	3+
Scholes et al. [33] (IPAQ-SF)	UK	CCV	sitting (PASBAQ) male	Pearson r = 0.23	3+
			sitting (PASBAQ) female	Pearson r = 0.29	3+

Notes: * Spearman ρ, unless otherwise labelled; AT—Austria; B—Belgium; DE—Germany; ES—Spain; FI—Finland; FR—France; LT—Lithuania; NL—The Netherlands; PT—Portugal; SE—Sweden; SI—Slovenia; UK—United Kingdom; CRV—criterion validity; CCV—concurrent validity; SB—sedentary behaviour; IDEAA— Intelligent Device for Energy Expenditure and Activity; PAQ24—Physical Activity Questionnaire for 24 h; PASBAQ—Physical Activity and Sedentary Behaviour Assessment Questionnaire.

**Table 5 ijerph-18-04602-t005:** Summary results for test–retest reliability, concurrent validity and criterion validity across all included studies.

Measurement Characteristic	Sample			Population Effect				Egger’s Bias Test			Heterogeneity		
	*N* (k)	k	*n*	Unweighted Mean	Weighted Mean	95% CI	80% CRI	Bias	95% CI	*p*	I^2^ (%)	Q	*p*
Reliability	5 [20,21,32,34,47]	36	6719	0.66	0.59	0.55 to 0.63	0.42 to 0.66	−0.98	−3.22 to 1.24	0.38	86.29	386.63	0.00
Concurrent validity	5 [20,25,33,34,50]	18	3074	0.72	0.55	0.42 to 0.68	0.20 to 0.90	−3.90	−11.26 to 3.46	0.31	96.47	481.49	0.00
Criterion validity	10 [20,21,31,32,34,44,45,46,48,49]	24	2164	0.22	0.23	0.19 to 0.27	0.19 to 0.25	0.89	−1.27 to 3.06	0.43	10.54	19.0	0.33

Notes: *N*—number of studies; k—number of associations for selected construct and measurement characteristics; *n*—number of participants; CI—confidence interval; CRI—credibility interval; I^2^—I index of heterogeneity; Q—chi-square test of heterogeneity.

**Table 6 ijerph-18-04602-t006:** Results of the risk-of-bias assessment.

Author (year)	Outcome	R	BC	BV	T	BM	VO	DA	RR	PC	Total
Bull et al. (2009) [25]	GPAQ +	0	0	1	1	0	1	0	1	0	4/9 (0.44)
Cámara et al. 2020 [48]	GPAQ −	0	0	1	0	0	1	0	1	0	3/9 (0.33)
Cleland et al. (2014) [49]	GPAQ −	1	0	1	1	0	1	1	1	1	7/9 (0.78)
Craig et al. (2003) [20]	IPAQ-SF *, +	0	0	0	0	0	1	0	1	0	2/9 (0.22)
Ekelund et al. (2005) [31]	IPAQ-SF −	1	0	0	1	0	1	0	0	0	3/9 (0.33)
Kalvenas et al. (2016) [32]	IPAQ-SF *	0	0	0	1	0	1	0	1	0	3/9 (0.33)
Kastelic et al. (2019) [45]	GPAQ −	0	0	0	1	0	1	0	1	0	3/9 (0.33)
Laeremans et a.l (2016) [46]	GPAQ −	0	0	1	1	0	1	0	0	0	3/9 (0.33)
Novak et al. (2020) [50]	GPAQ +	0	0	0	1	0	1	0	1	1	4/9 (0.44)
Rivière et al. (2016) [34]	GPAQ *, +, −	0	0	0	1	0	1	0	1	1	4/9 (0.44)
Rosenberger et al. (2008) [21]	IPAQ-SF *	0	0	0	1	0	1	0	1	0	3/9 (0.33)
Rudolf et al. (2020) [44]	GPAQ −	0	0	0	1	0	1	1	1	0	4/9 (0.44)
Rütten et al. (2003) [47]	IPAQ-SF *	1	0	1	1	0	1	0	1	1	6/9 (0.67)
Scholes et al. (2016) [33]	IPAQ-SF +	1	0	0	1	0	1	0	1	0	4/9 (0.44)
AVERAGE OF ALL STUDIES		0.29	0.00	0.36	0.86	0.00	1.00	0.14	0.86	0.29	0.42

R—randomisation; BC—Baseline comparable; BV—Baseline values accounted for in analyses; T—timing; BM—blinding of measures; VO—validated outcome measures; DA—dropout analysis; RR—reporting of results; PC—power calculation; Total—total score of the risk of bias (decimal format); * outcome for test–retest reliability; + outcome for concurrent validity; − outcome for criterion validity; 0—does not meet the criterion; 1— meets the criterion.

## Data Availability

The data presented in this study are available in this paper.

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
