# Peer review of "Validity and Reliability of IPAQ-SF and GPAQ for Assessing Sedentary Behaviour in Adults in the European Union: A Systematic Review and Meta-Analysis"

_ijerph, 2021, doi:10.3390/ijerph18094602_

Round 1

Reviewer 1 Report

The authors conducted a systematic review and meta-analysis to examine the validity and reliability of various physical activity questionnaires. They found moderate to high reliability, and mixed results with respect to validity, and also highlighted some concerns regarding the quality of the included studies. I have a few comments on the current work, including major concerns about the transparency of the methods.

Overall manuscript:

  1. In the abstract and indeed throughout the manuscript, the authors should refrain from using acronyms as far as possible as it is not only confusing but also distracting to the reader. Additionally, the authors fail to define what MET stands for before the first use in line 33, and fail to define what ES stands for before the first use in line 155.
  2. The manuscript would benefit from thorough proofreading. For example, there is a missing fullstop at the end of line 42, and ResearchGate is misspelled in line 113. The presence or absence of a space before and after mathematical symbols (e.g., “=” vs. “ = “) is inconsistent. The number of decimal places in Table 3 “Results” column is inconsistent.

Introduction:

  1. In lines 54–70, the authors mention two previous meta-analyses on a similar/broader topic that the current meta-analysis aims to address. It is unclear why there is a need for the current meta-analysis or what the contribution of the current meta-analysis is if these two previous works already exist. The authors should be specific about the purpose/strength of their work in comparison with the two existing works.

Methods:

  1. What is the reason for excluding unpublished or grey literature (e.g., dissertations)?
  2. The authors mention that “The included studies needed to report modes of administration, translation protocols and coefficients for reliability, concurrent validity, and criterion validity.” in lines 137–139. What did the authors do if a record met all other criteria but not this last criterion? Were the original authors contacted for the information or was the record immediately excluded? If the latter, why was there no attempt made to contact the original authors?
  3. Who conducted the eligibility check/screening procedure? It is mentioned in lines 149–150 but should instead be mentioned under Section 2.2 so that readers are not left wondering.
  4. What was the interrater agreement/interrater reliability for the risk of bias assessment? How were discrepancies resolved?
  5. What data were extracted exactly? In Section 2.4 the authors only mention who performed the extraction, but not what was extracted.
  6. The forest plot tool (“DistillerSR Forest Plot Generator”) should be cited.
  7. The PRISMA flowchart is incomplete (e.g., in the “Records excluded” box, the last line seems to be cut off).

Results:

  1. In Table 6, it is unclear what 0 and 1 refer to. Is it 0=bad and 1=good or the other way? A table note would be sufficient.

Reviewer 2 Report

This manuscript, entitled "Validity and reliability of IPAQ-SF and GPAQ for assessing sedentary behavior in adults in the European Union: a systematic review and meta-analysis.", Aimed to critically review, compile, and assess the reliability, criterion validity, and construct validity of the single-item SB questions within national language versions of the most commonly used international PAQs in the EU.

The study is interesting, but it has some limitations that need to be resolved.
Although it presented in the title and text that a meta-analysis was carried out, I believe that it was not carried out. Therefore, it is necessary to review this issue in the manuscript. Furthermore, Figure 1. does not represent a meta-analysis forest plots. It is important to review English.

Title: Was a meta-analysis really carried out? I did not identify a meta-analysis in the study. Explain this better.

Abstract Standardize the objective of the abstract of the objective described at the end of the introduction.

Keywords: Lines 25 and 26. Review including “Mesh Terms” from available at https://www.ncbi.nlm.nih.gov/mesh/

INTRODUCTION
Insert a period at the end of the first and second paragraph of the introduction.
Line 37: Insert reference after: “their waking hours sitting are classified as sedentary”.

Lines 43 and 44: Insert information about methods that can be classified as objective and subjective in the assessment of sedentary behavior.
Insert strengths and limitations with the use of physical activity assessment questionnaires.
Lines 102 to 105. Standardize information on the objectives available in the abstract and introduction.

METHODS
ResearhGate databases is not a database commonly used to include articles in a systematic review.
I suggest adding another database, either Embase or Scopus.
Insert the search strategy performed in PubMed.
Figure 1. Flowchart needs to be corrected. It is important that in figure 1 and the results text it is described how many articles were identified in each revised database.
Line 155: The use of forest plots has been described in statistical analysis. However, results from “forest plots” are not presented in the manuscript.

RESULTS
Explain better in the methodology and the results on Figure 2. The image does not represent a “Forest plot” of meta-analysis calculations. What kind of graph was used?

Table 5: In “N (k)” insert the study references.

Round 2

Reviewer 1 Report

The authors have sufficiently addressed my concerns and comments. I appreciate their effort and hard work.

Reviewer 2 Report

All my comments were correctly addressed bu the authors.